# The Gain-of-Function Mutation, *OsSpl26,* Positively Regulates Plant Immunity in Rice

**DOI:** 10.3390/ijms232214168

**Published:** 2022-11-16

**Authors:** Huihui Shang, Panpan Li, Xiaobo Zhang, Xia Xu, Junyi Gong, Shihua Yang, Yuqing He, Jian-Li Wu

**Affiliations:** 1State Key Laboratory of Rice Biology, China National Rice Research Institute, Hangzhou 310006, China; 2College of Life Science and Technology, Huazhong Agricultural University, Wuhan 430070, China

**Keywords:** rice, spotted-leaf mutant, *Osspl26*, reactive oxygen species, bacterial blight, cell death

## Abstract

Rice spotted-leaf mutants are ideal materials to study the molecular mechanism underlying programmed cell death and disease resistance in plants. *LOC_Os07g04820* has previously been identified as the candidate gene responsible for the spotted-leaf phenotype in rice *Spotted-leaf 26* (*Spl26)* mutant. Here, we cloned and validated that *LOC_Os07g04820* is the locus controlling the spotted-leaf phenotype of *Spl26* by reverse functional complementation and CRISPR/Cas9-mediated knockout of the mutant allele. The recessive wild-type *spl26* allele (*Oryza sativa spotted-leaf 26*, *Osspl26*) is highly conservative in grass species and encodes a putative G-type lectin S-receptor-like serine/threonine protein kinase with 444 amino acid residuals. OsSPL26 localizes to the plasma membrane and can be detected constitutively in roots, stems, leaves, sheaths and panicles. The single base substitution from T to A at position 293 leads to phenylalanine/tyrosine replacement at position 98 in the encoded protein in the mutant and induces excessive accumulation of H_2_O_2_, leading to oxidative damage to cells, and finally, formation of the spotted-leaf phenotype in *Spl26*. The formation of lesions not only affects the growth and development of the plants but also activates the defense response and enhances the resistance to the bacterial blight pathogen, *Xanthomonas oryzae* pv. *oryzae*. Our results indicate that the gain-of-function by the mutant allele *OsSpl26* positively regulates cell death and immunity in rice.

## 1. Introduction

Plant innate immunity is a sophisticated strategy developed in the long-term coevolution of plants and microorganisms to resist most pathogens. It mainly includes two levels, pathogen-associated molecular pattern (PAMP)-triggered immunity (PTI) and effector-triggered immunity (ETI) [1,2,3]. Plant cell death is genetically programmed and is often induced in the process of plants resisting pathogen infection through the immune system. The main factors causing programmed cell death (PCD) include reactive oxygen species (ROS), proteases and mitochondrial functional changes [4]. The hypersensitive response (HR) belongs to a type of PCD and is triggered by the activation of immune receptors after the invading pathogen is recognized at the site of infection. It is characterized by rapid cell death at/around the infected sites, helping plants to prevent further invasion by the pathogens [5,6,7,8].

Most spotted-leaf mutants (lesion mimic mutants) can spontaneously form HR-like necrotic lesions without biotic and abiotic stresses, resulting in enhanced plant resistance to pathogens. These mutants exist widely in various plant species and are closely associated with PCD and plant disease resistance [9,10,11,12,13,14]. The rice lesion mimics mutant *oscul3a* and shows the accumulation of ROS, up-regulated expression of disease-related genes and significantly increased resistance to *Magnaporthe oryzae* and *Xanthomonas oryzae* pv. *oryzae* (*Xoo*) [15]. The rice spotted-leaf mutant *spl24* exhibits enhanced resistance to multiple races of *Xoo* with elevated expression of pathogenesis-related genes [16]. Insight into the function of spotted-leaf genes plays a critical role in dissecting the signal regulatory network of PCD and characterizing the molecular mechanism of disease resistance in plants [17].

Protein phosphorylation catalyzed by protein kinases is an important mechanism for regulating signal transduction in plants. Studies have shown that protein kinases are involved in a series of life processes, such as plant growth and development, plant stress resistance and various signal transduction [18,19]. Receptor-like protein kinases (RLKs) belong to a subfamily of protein kinases, which turn on or off downstream target proteins through phosphorylation or dephosphorylation of the intracellular kinase region and transform extracellular signals into cytoplasmic signals [20]. RLKs are generally composed of a single transmembrane peptide chain, which generally consists of three regions, including an extracellular domain, a transmembrane region and an intracellular protein kinase region. It has been shown that RLKs are involved in many physiological and biochemical reactions and play a fundamental role in plant disease resistance [20,21,22,23]. The rice *Xa21* encodes a plant receptor-like kinase that confers broad-spectrum resistance to multiple races of *Xoo* [24]. The Arabidopsis receptor-like kinase 902 (AtRLK902) plays an important role in resisting the infection of the bacterial pathogen *Pseudomonas syringae* [25]. The rice *SPL36* encodes a receptor-like protein kinase, which regulates the defense response of rice, and the loss-of-function mutant *spl36* exhibits an enhanced level of resistance to rice bacterial blight strain HM73 [26].

We previously identified a novel gain-of-function spotted-leaf mutant *Spl26* from an ethylmethylsulfone (EMS)-induced rice cultivar IR64. The spotted-leaf phenotype is controlled by a dominant nuclear gene located in the short arm of chromosome 7 [27]. Here, we show that the rice OsSPL26, a putative G-type lectin S-receptor-like serine/threonine protein kinase encoded by *LOC_Os07g04820*, controls the spotted-leaf phenotype. The gain-of-function mutation by the mutant allele *OsSpl26* causes the accumulation of H_2_O_2_, leading to HR-like cell death, and positively regulates enhanced resistance to *Xoo* in rice.

## 2. Results

### 2.1. OsSpl26 Controls the Spotted-Leaf Phenotype of Spl26

In the previous study, we mapped the mutation responsible for the spotted-leaf phenotype of *Spl26* in a 305 kb region in chromosome 7 [27]. Here we carried out whole genome sequencing on *Spl26* and the wild-type IR64. Sequence alignment indicated that a single base substitution (A/T) at position 293 of *LOC_Os07g04820* was identified in *Spl26*. To confirm whether the single base substitution in the mutant allele was responsible for the *Spl26* phenotype, we performed a reverse functional complementation analysis. Since the *Spl26* mutation is dominant, we cannot introduce the wild-type allele (*Osspl26*) to complement the mutant phenotype. Instead, we used the mutant allele (*OsSpl26*) to reproduce the spotted-leaf phenotype. Therefore, we introduced the construct p1300-SPL26-C containing the promoter and genomic coding region of *OsSpl26* into the calli of Nipponbare (NPB). A total of 26 transgenic lines were obtained, of which four were positive with bimodal mutation sites; however, only two out of four lines (CP-9 and CP-18) showed the spotted-leaf phenotype similar to that of *Spl26* (Figure 1A–C). Therefore, CP-9 and CP-18 were considered positive reverse complementary lines. Compared with the wild-type NPB, the chlorophyll contents of the reverse complementary lines were significantly degraded (Figure 1D). The reverse complementary lines also exhibited a poorer performance in major agronomic traits compared with the wild-type (Appendix A). We also carried out CRISPR/Cas9-mediated editing on *LOC_Os07g04820* using *Spl26*-derived mature embryogenic calli. A total of 18 independent knockout lines were obtained, and all of them did not show any spotted-leaf phenotype. Three independent knockout lines (Cr-7, Cr-8 and Cr-9) with similar growth status to IR64 were selected for further experiments (Figure 1E–G). The chlorophyll contents of Cr-7, Cr-8 and Cr-9 were similar to that of the wild-type and significantly higher than that of *Spl26* (Figure 1H). Although the agronomic traits of knockout lines did not fully recover to the wild-type level, they were significantly higher than that of *Spl26* (Figure 1I–L). Taken together, these results indicate that *LOC_Os07g04820* is the target locus of *Osspl26* and responsible for the spotted-leaf phenotype in *Spl26*.

### 2.2. Phenylalanine Is Highly Conserved in Plant SPL26

According to the annotation given by the Rice Genome Annotation Project (http://rice.uga.edu/, accessed on 7 January 2019), *Osspl26* encodes a putative protein kinase. The coding sequence of *Osspl26* consists of 1335 nucleotides, encoding a putative protein containing 444 amino acid residuals with a molecular weight of 48.4 kDa. The secondary structure of OsSPL26 contains a transmembrane domain (20–42 aa) and a kinase domain (109–385 aa) as predicted in the SMART (http://smart.embl-heidelberg.de/, accessed on 17 July 2019).

According to the amino acid sequence of OsSPL26, we found 11 plant homologues in NCBI by BLASTP. The results show that SPL26 is conserved in plant species, including *Oryza sativa*, *Triticum turgidum*, *Hordeum vulgare*, *Setaria italica*, *Zea mays*, *Sorghum bicolor*, *Elaeis guineensis*, *Cocos nycifera*, *Glycine max*, *Arabidopsis thaliana* and *Nicotiana tabacum*. Among these homologues, OsSPL26 has the highest similarity with the homologue of *Hordeum vulgare* (79%) and the lowest similarity with the homologue of *Arabidopsis thaliana* (44%). To compare the evolutionary relationship between SPL26 homologues, a phylogenetic tree of SPL26 was constructed. As shown in Figure 2, the six monocot homologues of SPL26 form a single group containing *Oryza sativa*, *Hordeum vulgare*, *Triticum turgidum*, *Setaria italica*, *Zea mays* and *Sorghum bicolor*, indicating that SPL26 may have conserved functions in grass species (Figure 2B). Moreover, phenylalanine (F) at position 98 in OsSPL26 is highly conserved among all the SPL26 (Figure 2A, red arrow), and its substitution by tyrosine likely alters its function and subsequently affects plant immunity, growth and development.

### 2.3. Osspl26 Is Constitutively Expressed and OsSPL26 Localizes to the Plasma Membrane

To investigate the spatial and temporal expression pattern of *Osspl26*, the total RNA of various tissues from the wild-type IR64 in different developmental stages was extracted and synthesized into cDNA, and then the relative expression was detected by quantitative reverse transcription-polymerase chain reaction (qRT-PCR). The results demonstrated that *Osspl26* was expressed in all examined tissues, including the roots, shoots, stems, leaves, sheaths and panicles (Figure 3A). Among the tissues, it had the lowest expression in the stems (Figure 3A). We also introduced the construct p1381Z-Gus containing the promoter of *Osspl26* to drive the expression of the *β-glucuronidase* (*GUS*) reporter gene in Nipponbare to further detect the spatial and temporal expression pattern of *Osspl26*. Our results indicated that the GUS signals were detected in all tissues tested (Figure 3B), which is consistent with the results of qRT-PCR. The results demonstrate that *Osspl26* is a constitutively expressed gene in rice.

We then introduced the construct p580-sub and the control vector into the rice protoplasts to determine the subcellular localization of OsSPL26, respectively. The red dye FM4-64 was used as a marker of plasma membrane localization. The results showed that the OsSPL26-GFP (green fluorescence protein) fusion proteins localized to the plasma membrane compared to the cytosol-localized control GFP (Figure 4). To detect the effect of gene mutation on protein localization, OsSPL26^F98Y^-GFP fusion proteins were expressed in rice protoplasts. It was found that OsSPL26^F98Y^-GFP proteins also localized to the plasma membrane, indicating that the 98th amino acid substitution (F/Y) in OsSPL26 did not affect its subcellular localization (Figure 4), indirectly supporting that OsSPL26^F98Y^ is a functional protein.

### 2.4. OsSpl26 Causes ROS Accumulation in Spl26

Previously we found the abnormality of the reactive oxygen species (ROS) scavenging system, resulting in the large accumulation of H_2_O_2_ and cell death in *Spl26* [27]. To determine whether the accumulation of H_2_O_2_ in *Spl26* was really caused by the mutation of *Osspl26*, the mutant allele *OsSpl26* was introduced into Nipponbare. We found that the H_2_O_2_ contents of the two reverse complementary lines (CP-9 and CP-18) were 1.56 times and 3.18 times significantly higher than that of Nipponbare, respectively (Figure 5A). In addition, compared with Nipponbare, the activities of catalase (CAT), ascorbate peroxidase (APX) and superoxide dismutase (SOD) was largely similar to that of Nipponbare (Figure 5B–D), while the activity of peroxidase (POD) was increased significantly in the reverse complementary lines (Figure 5E). Furthermore, when *OsSpl26* was knocked out, the H_2_O_2_ contents of the knockout lines were significantly reduced compared with *Spl26* (Figure 5F), while the activities of CAT and APX increased apparently compared with *Spl26* (Figure 5G,H). The activities of SOD and POD were similar to that of IR64 but significantly lower than that of *Spl26* (Figure 5I,J). These results indicated that the gain-of-function by *OsSpl26* caused the disorder of the ROS scavenging system, leading to the excessive accumulation of H_2_O_2_ in *Spl26*.

It has been shown that an appropriate level of ROS, as signal molecules, is critical to plant growth and development. However, a high concentration of ROS will cause oxidative damage to lipids, proteins and DNA in plant cells, thus inducing cell death and lesion formation [17]. To validate, we measured the content of soluble proteins and malondialdehyde (MDA), the product of membrane lipid peroxidation in the mutant. In the reverse complementary lines (Nipponbare background), the MDA content increased markedly, and the soluble protein contents decreased significantly compared with the wild-type (Figure 5K,L). In the knockout lines, the contents of MDA and soluble proteins recovered to similar levels of the wild-type (Figure 5M,N). The results further indicated that the cell death and lesion formation were likely caused by excessive accumulation of ROS in *Spl26*. In conclusion, the above results demonstrate that the gain-of-function by *OsSpl26* positively regulates the production of H_2_O_2_, triggers cell damage and death and leads to lesion formation in *Spl26*.

### 2.5. Mutation of OsSpl26 Positively Regulates Immunity in Rice

The rice spotted-leaf mutants usually exhibit enhanced resistance to pathogen infection with up-regulated expression of defense response genes [28]. Previous studies showed that *Spl26* significantly increased the resistance to two *Xoo* races, PXO86 and PXO99 [27]. To determine that the enhanced resistance was due to the mutation of *Osspl26*, the transgenic lines were inoculated with PXO99. In the reverse complementary lines harboring *OsSpl26*, the lesion length and disease index were significantly lower than those of the wild-type NPB (Figure 6A–C). In contrast, the lesion length and disease index of the knockout lines were similar to those of IR64 and were significantly longer and higher than those of *Spl26*, respectively (Figure 6D–F). The results indicate that the mutant allele *OsSpl26* is the factor responsible for the enhancement of bacterial blight resistance.

To further determine the enhanced disease resistance associated with the mutation of *Osspl26*, the expression of a set of representative defense response genes [27] in transgenic lines was evaluated by qRT-PCR. The results showed that the expression levels of the defense response genes were markedly up-regulated in the reverse complementary lines compared to NPB (Figure 6G). In contrast, the expression levels of these genes were largely down-regulated in the knockout lines compared with *Spl26* (Figure 6H). The results showed that the gain-of-function mutation could activate the defense response in rice. Taken together, these results indicate that *OsSpl26* activates a number of defense response genes, leading to enhanced resistance to *Xoo* in rice.

## 3. Discussion

The spotted-leaf mutant usually develops HR-like symptoms; thus, it is an ideal material to characterize the molecular mechanism of programmed cell death and disease resistance. Unlike many recessive mutations, the mutation of *Spl26* is homozygous dominant lethal (*OsSpl26*/*OsSpl26*) and can only survive in the heterozygous form (*OsSpl26*/*Osspl26*) that displays reddish-brown necrotic lesions approximately 3 weeks after sowing and the leaf edges wilts and dries up gradually in later growing stages [27]. In the present study, we isolated the target locus *LOC_Os07g04820* and proved that it is the right gene responsible for the spotted-leaf phenotype in *Spl26*. Due to the recalcitrance nature of the indica genotype of IR64 during transformation, we introduced the mutant allele *OsSpl26* into the japonica rice Nipponbare, and the positive transformants exhibited a spotted-leaf phenotype similar to *Spl26* (Figure 1A,B), although two positive transformants did not appear the spotted-leaf phenotype due to unknown reasons. In addition, the knockout lines did not show the spotted-leaf phenotype as expected (Figure 1F,G). Like many spotted-leaf mutants, the mutation of *Spl26* imposed a severe negative impact on the performance of major agronomic traits, including yield and yield components [16,27,29,30,31,32]. In this study, the mutant and the reverse complementary lines had lower chlorophyll contents compared with their corresponding wild-types (Figure 1D); in contrast, the knockout lines and IR64 had a similar chlorophyll level (Figure 1H). The 1000-grain weight and seed-setting rate of the reverse complementary lines were lower than those of the wild-type NPB. In contrast, the 1000-grain weight and seed-setting rate were increased in the knockout lines compared with *Spl26* (Appendix A and Figure 1I–L). We speculate that the formation of necrotic lesions on the leaves affects the ability of photosynthesis in *Spl26*, leading to poor performance of growth and development. Our results demonstrate that the functional allele *OsSpl26* results in lesion formation and degrades the mutant performance of major agronomic traits.

It has been shown that lesion formation in spotted-leaf mutants usually results in enhanced disease resistance [16,28,33,34]. Mostly, lesion formation directly results from excessive accumulation of a certain type of ROS, which subsequently activates the elevated expression of defense response genes, leading to enhanced resistance to disease pathogens [17,28]. In this study, the mutant allele *OsSpl26* also induced the lesion formation and up-regulated expression of multiple defense response genes and the enhanced resistance of plants to bacterial blight pathogens. Firstly, the lesion formation is associated with the high level of H_2_O_2_ accumulation in *Spl26*. In fact, the H_2_O_2_ level also sharply increased when *OsSpl26* was introduced into the reverse complementary lines, while it was markedly reduced when *OsSpl26* was knocked out. Secondly, as expected, the excessive accumulation of H_2_O_2_ is associated with the disorder of the intracellular ROS scavenging system in *Spl26* (Figure 5). Consequently, the mutant and the reverse complementary lines harboring the mutant allele exhibit activated expression of multiple defense genes and enhanced resistance to the virulence race of PXO99, while the knockout lines show a reduced level of resistance to PXO99 with decreased expression levels of typical defense response genes (Figure 6). Taken together, *OsSpl26* induces excessive accumulation of H_2_O_2_, which leads to oxidative damage to cells and, finally, the formation of the spotted-leaf phenotype. Therefore, on the one hand, the mutation impedes plant growth and development; on the other hand, it activates the defense response and enhances disease resistance.

According to the information given by the Rice Genome Annotation Project, *Osspl26* encodes a putative protein kinase. Then, we compared a set of SPL26 homologues in plants to determine the potential relationship among them. The BLASTP results show that OsSPL26 could be a G-type lectin S-receptor-like serine/threonine protein kinase. However, the secondary structure of OsSPL26 predicted by SMART shows that OsSPL26 contains a transmembrane domain and a kinase domain, lacking the G-type lectin domain or S domain. Thus, OsSPL26 is likely a novel receptor-like protein kinase with an untypical 19 aa extracellular domain [35]. In addition, plant receptor-like kinases are usually located in the plasma membrane and are able to perceive and respond to extracellular signals [36]. As expected, OsSPL26-GFP localizes to the plasma membrane in rice (Figure 4). Nevertheless, we need additional evidence to determine whether the extracellular domain is able to recognize outside signal molecules such as PAMPs or if it just acts as a signal peptide for membrane localization. Currently, we are carrying out further research to identify the potential proteins that interact with OsSPL26.

The mutation of a single nucleotide (T/A) in *Osspl26* results in the substitution of phenylalanine (F) for tyrosine (Y). The amino acid sequence alignment shows that phenylalanine (F) at position 98 is an extremely conservative site in the SPL26 homologues (Figure 2A). This makes us wonder why the phenylalanine/tyrosine substitution is a dominant gain-of-function mutation. Tyrosine phosphorylation involves cellular communication in animals [37]. Plant receptor-like kinases are traditionally classified as serine/threonine kinases, but some of them also show the ability to phosphorylate tyrosine. The *Botrytis*-induced kinase 1 (BIK1) in Arabidopsis is a classical serine/threonine kinase, which is autophosphorylated and phosphorylated by BAK1 (brassinosteroid insensitive 1-associated kinase 1) at multiple tyrosine residues in response to plant immune signals [38]. The Arabidopsis receptor kinase EF-TU receptor (EFR) is activated by the phosphorylation of tyrosine residues as well. The phosphorylation of a single tyrosine residue, Y836, is necessary to activate EFR and immunity to plant pathogens [39]. Liu et al. show that a plant receptor kinase and phosphatase in Arabidopsis coordinate the Tyr phosphorylation cycle of CERK1 (chitin elicitor receptor kinase 1) to regulate its activation in response to the fungal chitin [40]. Thus, tyrosine phosphorylation of OsSPL26 could play a crucial role in plant immunity in rice. In addition, we identified a potential interaction between OsSPL26 and a ribulose bisphosphate carboxylase large chain precursor (OsrbcL1, data unpublished). It has been known that Rubisco, acting as an oxygenase, participates in the photorespiratory pathway, which is a major source of H_2_O_2_ in the cells [41,42]. We postulate that OsrbcL1 might compete with OsrbcL in the assembly of Rubisco, leading to its multifunction or the OsSPL26/OsrbcL1 interaction impeding the final processing of OsrbcL1 and producing an excessive amount of H_2_O_2_, resulting in lesion formation and cell death in *Spl26*. Although the fact that the gain-of-function of protein kinase OsSPL26 can activate defense response and enhance disease resistance, whether it is caused by tyrosine phosphorylation or OsSPL26/OsrbcL1-mediated H_2_O_2_ over-production needs to be further characterized.

## 4. Materials and Methods

### 4.1. Plant Materials and Growth Conditions

The spotted-leaf mutant *Spl26* was obtained from indica rice (*Oryza sativa*) IR64 by ethyl methylsulfonate (EMS) chemical mutagenesis. The *Spl26* mutant, wild-type IR64, and Nipponbare were planted in the paddy field under natural summer conditions at the China National Rice Research Institute (CNRRI), Hangzhou, China. All transgenic plants were grown in the greenhouse at CNRRI under normal water and fertilizer management.

### 4.2. Agronomic Trait Evaluation

At full maturity, the main agronomic traits, including plant height, panicle length, 1000-grain weight and seed-setting rate, were evaluated from three randomly selected plants from IR64, *Spl26* and the T_1_ knockout lines. Means of three replicates were used for analysis. Only a single plant was evaluated for the agronomic traits of Nipponbare and the reverse complementary lines.

### 4.3. Vector Construction and Transformation

For the reverse complementation test, the mutant allele *OsSpl26* containing a 2692 bp sequence upstream of the transcription start site, a 1741 bp full-length genomic DNA and a 1500 bp sequence downstream of the termination site was cloned into the vector pCAMBIA1300 to form a new construct p1300-SPL26-C. This new construct was transformed into embryogenic calli derived from Nipponbare mature seeds by an *Agrobacterium tumefaciens*-mediated method [43]. For the CRISPR/Cas9-mediated knockout, two target sites were chosen simultaneously. Target one covered the sequence from −5 bp to 15 bp, and target two covered the sequence from 515 bp to 534 bp. The *OsSpl26* knockout construct, pSPL26-Cr, was then generated according to the previous description [44]. This construct was then transformed into embryogenic calli induced from the mutant *Spl26* (heterozygote) mature seeds by an *Agrobacterium tumefaciens*-mediated method [43]. For the GUS transient assay, the 2692 bp promoter region of *Osspl26* was cloned into plasmid pCAMBIA1381Z to form a new construct p1381Z-Gus which then was transformed into the embryogenic calli induced from Nipponbare (NPB) mature seeds. Different tissues of transgenic plants were stained with the GUS staining kit according to the manufacturer’s instructions (Coolaber, SL7160). The pictures were recorded using a Leica S8APO stereo microscope. For subcellular localization, the 1335 bp full-length coding sequence of *Osspl26* (without stop codon) was cloned into PAN580 vector to form a new construct p580-sub, and then p580-sub and the control vector were transformed into the rice protoplast derived from Nipponbare seedlings, respectively, according to the previous method [45]. The red dye FM4-64 is used as a marker of plasma membrane localization. The fluorescence signal was observed by LMS700 confocal laser scanning microscope (Carl Zeiss, Oberkochen, Germany). Primers used for vector construction are listed in Appendix A.

### 4.4. Measurement of Physiological Parameters

Total chlorophylls of the reverse complementary lines, knockout lines, *Spl26*, Nipponbare and IR64 were extracted from the uppermost fresh leaves of the plants at the tillering stage. After the leaves were cut into small sections, a 0.01 g sample was immersed in 1 mL 95% ethanol in the dark for 48 h. Total chlorophyll content was determined by measuring A_652_ of samples solution using a SpectraMax i3x (Molecular Devices, San Jose, CA, USA). Total chlorophyll content (mg/g FW) is calculated as (V × A_652_)/(34.5 × m). V (mL) indicates sample volume in 95% ethanol, and m (g) indicates leaf sample weight.

Fresh leaf samples from IR64, *Spl26*, knockout lines, Nipponbare and reverse complementary lines at the tillering stage were used for the determination of the following parameters. The contents of H_2_O_2_, MDA and soluble proteins and the activities of ROS scavenging enzymes, including catalase (CAT), superoxide dismutase (SOD), peroxidase (POD) and ascorbate peroxidase (APX) were determined by using the kits according to the manufacturer’s instructions (Nanjing Jiancheng Bioengineering Institute, Nanjing, China). Means from three replicates were used for analysis.

### 4.5. RNA Extraction and qRT-PCR

The total RNA was isolated using the NucleoZOL reagent according to the manufacturer’s instructions (MACHEREY-NAGEL, Düren, Germany). The first strand cDNA was reverse transcribed from 1 μg total RNA using the PrimeScript™ RT Master Mix (TaKaRa, Dalian, China). PowerUp™ SYBR™ Green Master Mix (Thermo Fisher Scientific, Waltham, MA, USA) was used for qRT-PCR and performed on a Thermal Cycle Dice Real Time System II according to the manufacturer’s instructions (TaKaRa, Dalian, China). The rice ubiquitin (*LOC_Os03g13170*) was used as an internal control. The data were analyzed by the 2^−ΔΔCt^ method, and means from three replicates were used for analysis. Primers used for qRT-PCR are listed in Appendix A.

### 4.6. Disease Evaluation

The *Xanthomonas oryzae* pv. *oryzae* strain PXO99 grown on WF-P solid medium (20.0 g sucrose, 5.0 g peptone, 0.5 g calcium nitrate, 0.82 g sodium phosphate, 0.2 g ferrous sulfate, 17.0 g agar in 1 L distilled water) was collected and diluted with sterile water and used for inoculation after adjusting the OD_600_ value to 1.0. At the maximum tillering stage, six fully expanded leaves from different plants were selected for inoculation using the leaf clipping method [28]. The lesion length of inoculated leaves was measured 14 days after inoculation using a transparent plastic ruler. The means of six replicates were used for analysis.

### 4.7. Phylogenetic Analysis

Blast analysis of the protein sequence of OsSPL26 was performed on the NCBI website (http://www.ncbi.nlm.nih.gov/, accessed on 16 February 2022) using the homologues of OsSPL26 from 11 plant species, including *Oryza sativa*, *Triticum turgidum*, *Hordeum vulgare*, *Setaria italica*, *Zea mays*, *Sorghum bicolor*, *Elaeis guineensis*, *Cocos nycifera*, *Glycine max*, *Arabidopsis thaliana* and *Nicotiana tabacum*. DNAMAN v7 (http://www.lynnon.com/, accessed on 16 February 2022) was used for multiple protein sequence alignment. The phylogenetic tree was constructed based on the neighbor-joining method provided by the MEGA v6 software (http://www. megasoftware.net/, accessed on 16 February 2022).

## Figures and Tables

**Figure 1 ijms-23-14168-f001:**
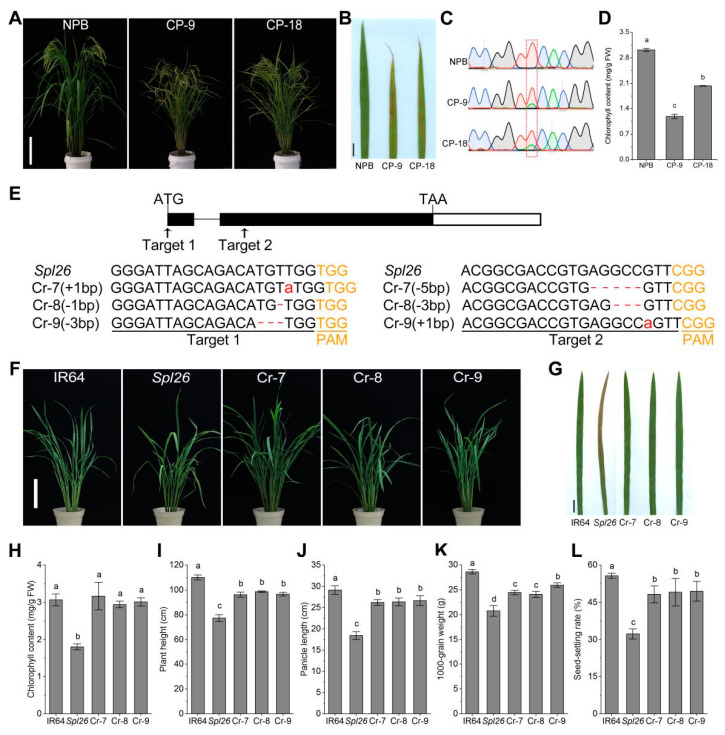
Reverse functional complementation and characterization of reverse complementary and knockout lines. (**A**) Phenotypes of wild-type Nipponbare (NPB) and reverse complementary lines CP-9, CP-18 at the heading stage. Bar = 17 cm; (**B**) Leaf phenotype of NPB and reverse complementary lines. Bar = 2 cm; (**C**) Genotype of NPB and reverse complementary lines CP-9, CP-18. The red box represents the mutation site, NPB has a single red peak (T), both CP-9 and CP-18 have a red/green double peak (T/A). Red peak indicates T, Blue peak indicates C, Green peak indicates A, Black peak indicates G; (**D**) Chlorophyll content of NPB and reverse complementary lines CP-9, CP-18 at the tillering stage; (**E**) CRISPR/Cas9-mediated mutations at the target sites in representative knockout lines (Cr-7, Cr-8 and Cr-9), small red letters indicate the corresponding base insertions, dot lines indicate deletions; (**F**,**G**) Phenotypes of IR64, *Spl26* and knockout lines Cr-7, Cr-8, Cr-9 at the tillering stage. (**F**), Bar = 18 cm. (**G**), Bar = 2 cm; (**H**) Chlorophyll content; (**I**–**L**) Major agronomic traits. Data are means ± SD (*n* = 3). Different letters indicate significant differences at *p* < 0.05 by Duncan’s multiple test.

**Figure 2 ijms-23-14168-f002:**
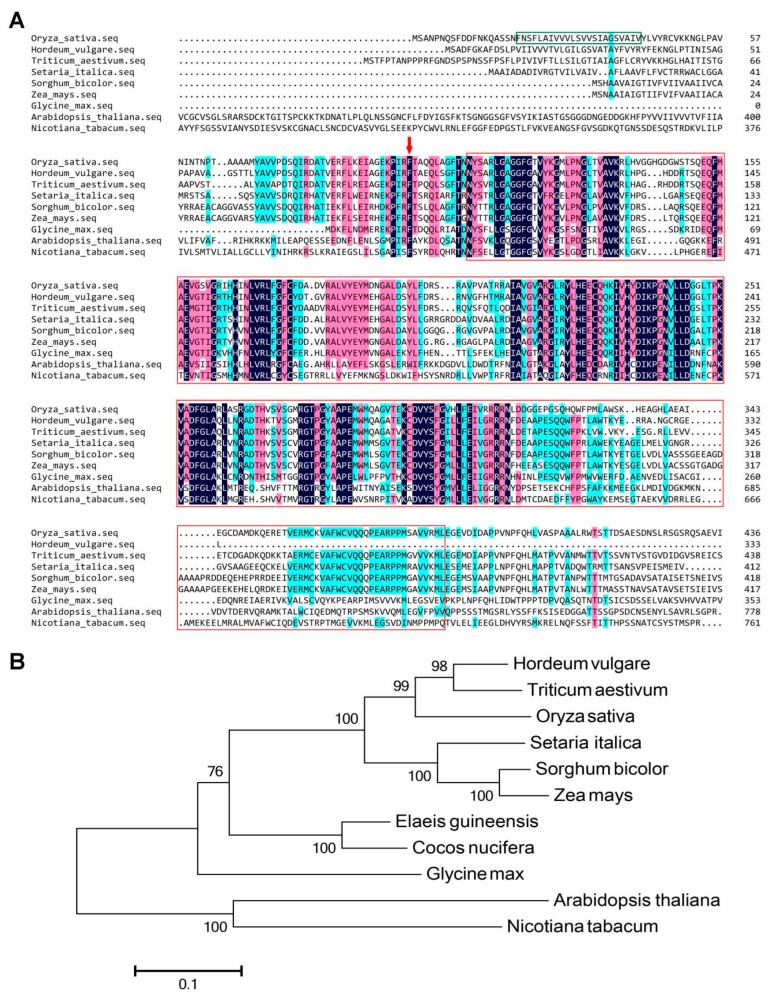
Sequence alignment and phylogenetic analysis of SPL26 homologues. (**A**) Partial sequence alignment of SPL26 homologues. The red arrow indicates the mutation site. The green box represents the transmembrane domain of OsSPL26. The red box represents the kinase domain of OsSPL26 and its homologues; (**B**) Phylogenetic tree of SPL26 proteins from 11 different species. The phylogenetic tree is constructed based on the neighbor-joining method using MEGA v6 software.

**Figure 3 ijms-23-14168-f003:**
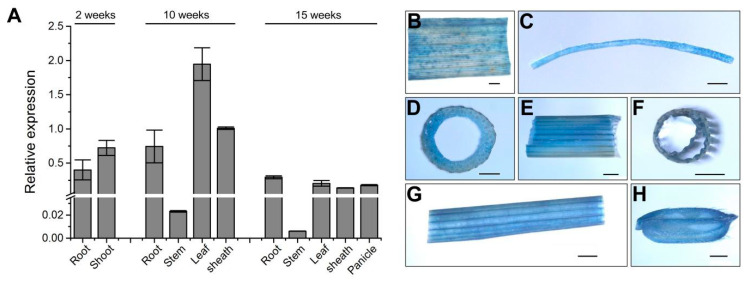
Relative expression of *Osspl26* and GUS transient expression assay. (**A**) The relative expression level of *Osspl26* in different tissues of IR64 at 2 weeks, 10 weeks and 15 weeks. Data are means ± SD (*n* = 3); (**B**–**H**) GUS signals detected in different tissues. Leaf (**B**), root (**C**), stem cross-section (**D**), stem (**E**), leaf sheath cross-section (**F**), leaf sheath (**G**), young spikelet (**H**). Bars = 1 mm.

**Figure 4 ijms-23-14168-f004:**
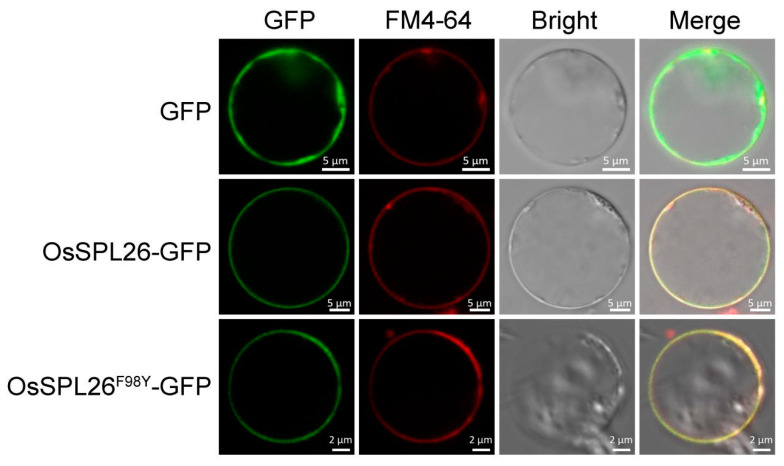
Subcellular location of OsSPL26-GFP in rice protoplasts. The red dye FM4-64 is used as a marker of plasma membrane localization. Upper panel: control GFP, Middle panel: OsSPL26 fused to GFP, Lower panel: OsSPL26^F98Y^ fused to GFP. GFP, Green fluorescence protein.

**Figure 5 ijms-23-14168-f005:**
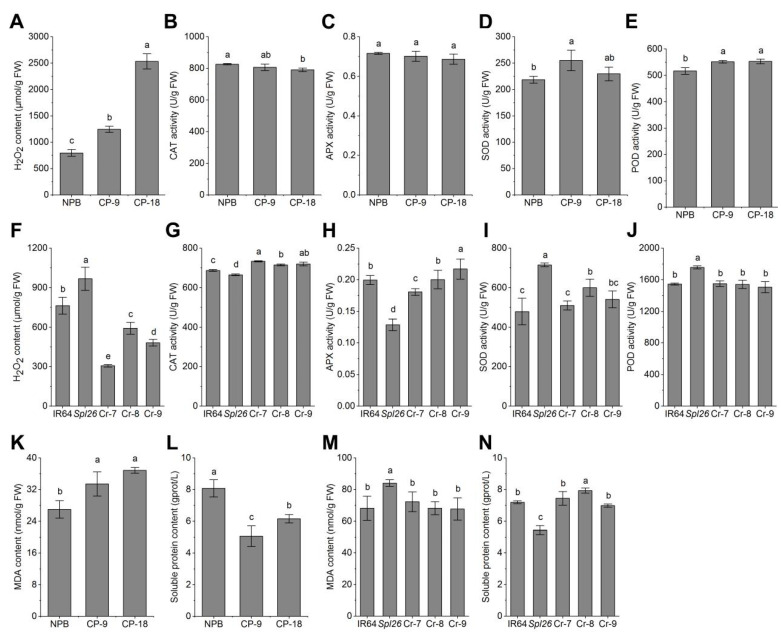
Physiological performance of transgenic lines and wild types at the tillering stage. (**A**,**F**) H_2_O_2_ content; (**B**,**G**) CAT activity; (**C**,**H**) APX activity; (**D**,**I**) SOD activity; (**E**,**J**) POD activity; (**K**,**M**) MDA content; (**L**,**N**) Soluble protein content. Data are means ± SD (*n* = 3), and different letters indicate significant differences at *p* < 0.05 by Duncan’s multiple test.

**Figure 6 ijms-23-14168-f006:**
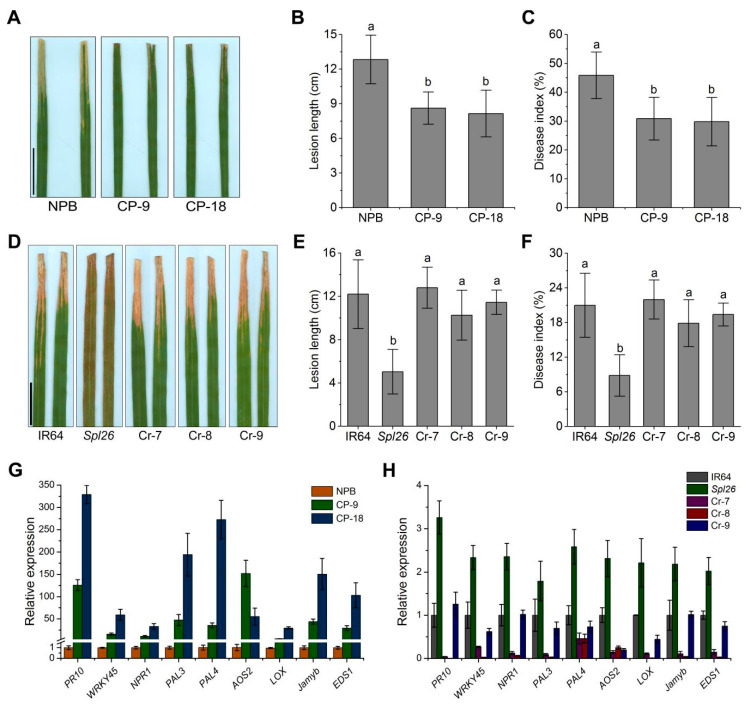
Disease evaluation and relative expression of defense response genes. (**A**,**D**) Responses of NPB, IR64, *Spl26*, and transgenic lines to race PXO99 at the tillering stage. Bars = 5 cm; (**B**,**E**) Lesion length; (**C**,**F**) Disease index. Data are means ± SD (*n* = 6), and different letters indicate significant differences at *p* < 0.05 by Duncan’s multiple test; (**G**,**H**) Expression levels of defense response genes in NPB, IR64, *Spl26* and transgenic lines. Data are means ± SD (*n* = 3).

## Data Availability

Data is contained within the article or Appendix A.

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
