# Peer review of "The Gain-of-Function Mutation, OsSpl26, Positively Regulates Plant Immunity in Rice"

_ijms, 2022, doi:10.3390/ijms232214168_

Round 1
Reviewer 1 Report
It is crucial to understand the molecular mechanisms underlying programmed cell death and disease resistance in plants. This research cloned and characterized the functions of a gene responsible for the spotted-leaf phenotype in a rice mutant Spl26 by using functional complementation and CRIPSPR/cas9-mediated knockout of the mutant allele. They confirmed the subcellular location of the encoded protein, showed the molecular mechanisms resulting in the formation of the lesion and impacts on plant growth, development, and defense against pathogen.
The authors designed elegant experiments by generating mutant allele line and knockout lines to show solid evidence for function characterization and validation, which well supports the conclusion the authors claimed. The results are well presented by several figures and the citations are well covered the background of this study, especially several classic reference papers about the plant innate immunity and programmed cell death.
However, I still have some concerns about the Figure 1, Figure 6 and the Method section for disease evaluation.
Figure 1, does Fig. 1A, 1B, 1F, and 1G show the disease phenotypes? It’s not clear to me what phenotypes they are. I would put the phenotypes of all lines at the same stages (i.e., heading in 1A and 1B, or tilling in 1F and 1G) for one comparison panel, including two WT, two complement lines, one mutant and three knockout lines.
Figure 6 and the Method section for disease evaluation, it is not clear to me the definition of lesion length and the method for measurement.
Reviewer 2 Report
The manuscript by Shang et al. "The gain of function mutation, OsSPL26, positively regulates plant immunity in rice" investigated Os07g04820 as a candidate gene for Spl26 phenotype. Initial results are good and well presented.
I have comments that need to be addressed
1. I am confused with gene/protein nomenclature. Author wrote SPL26 for mutant, while spl26 for wild type.
please adopt standard as below
SPL26: Wild type gene. Upper case italic
spl26: Mutant gene. lower case italic
SPL26: wild type protein. upper case non-italic
spl26: mutant protein. lower case non-italic
2. Authors are saying CP-9 and CP-18 lines as complemented lines. This is not correct. It is not complementing mutant phenotype, instead creating a mutant phenotype. It will be appropriate to mutant transgenic lines
3. Do NBP has wild type version of OsSPL26. Figure 1C showed mutant allele in NBP
4. Figure 1E showed sequence of IR26. I guess it should have been spl26 since they are knocking out mutant line
5. It is not clear what generation of CP-09/CP-18 and Cr-7/8/9 lines were used. It will good to have non-transgenic Cr-7/8/9 lines and evaluate phenotype
6. What percentage of mutation rate was in each of Cr-7, Cr-8 and Cr-9 lines.
7. Results 2.3 section, Figure 3. It is not clear whether authors used wild type or mutant version of OsSPL26 for their expression study. It will be important to compare in both version
Round 2
Reviewer 2 Report
Authors addressed all my comments.
Thanks